# A Review on Generation and Reactivity of the *N*-Heterocyclic Carbene-Bound Alkynyl Acyl Azolium Intermediates

**DOI:** 10.3390/molecules27227990

**Published:** 2022-11-17

**Authors:** Ziyang Dong, Chengming Jiang, Changgui Zhao

**Affiliations:** Key Laboratory of Radiopharmaceuticals, Ministry of Education, College of Chemistry, Beijing Normal University, Beijing 100875, China

**Keywords:** *N*-heterocyclic carbene, alkynyl acyl azolium, annulation, intermediate, mechanism

## Abstract

*N*-heterocyclic carbene (NHC) has been widely used as an organocatalyst for both umpolung and non-umpolung chemistry. Previous works mainly focus on species including Breslow intermediate, azolium enolate intermediate, homoenolate intermediate, alkenyl acyl azolium intermediate, etc. Notably, the NHC-bound alkynyl acyl azolium has emerged as an effective intermediate to access functionalized cyclic molecular skeleton until very recently. In this review, we summarized the generation and reactivity of the NHC-bound alkynyl acyl azolium intermediates, which covers the efforts and advances in the synthesis of achiral and axially chiral cyclic scaffolds via the NHC-bound alkynyl acyl azolium intermediates. In particular, the mechanism related to this intermediate is discussed in detail.

## 1. Introduction

Early in 1943, *N*-heterocyclic carbene (NHC) was discovered and applied to catalytic reactions in the form of coenzyme vitamin B1 [1]. Later in 1958, Breslow proposed an appropriate mechanism for a vitamin B1-catalyzed benzoin condensation reaction [2], which has inspired synthetic chemists to focus on NHCs in the field of catalytic organic reactions for decades. Due to its unique properties in organic chemical reaction processes, NHCs have been widely used as organometallic ligands as well as organocatalysts, owing to their extensive and diverse synthesis and versatility [3,4,5,6,7,8,9,10,11]. Generally, NHC-bound intermediates involving organocatalytic reactions are divided into the following types: Breslow intermediates, azolium enolate intermediates, homoenolate intermediates, azolium dienolate intermediates, and radical intermediates, as well as acyl azolium intermediates [12,13,14,15,16]. These intermediates have been explored for both umpolung and non-umpolung chemistry such as benzoin condensation, Stetter reaction, hydroacylation, [n + m] annulation, and so on.

Over recent decades, acyl azolium has represented a central reactive species for reaction designs in the modern era of NHC-based catalysis. Overall, acyl azolium intermediates can be categorized into the following types: alkyl acyl azoliums [17], alkenyl acyl azoliums [18,19,20], dienyl acyl azoliums [21,22,23], and alkynyl acyl azoliums [20] (Figure 1A). Among them, NHC-based alkynyl acyl azolium intermediates, initially identified independently by Chi [24], Du [25], and Wang [26] between 2017 and 2018, were less commonly studied. Although only a few reaction models involving the species have been developed, alkynyl acyl azolium intermediates have been recognized as a new NHC-bound specie for the discovery of new reactions. Until now, three procedures have been disclosed to produce alkynyl acyl azoliums intermediates according to their precursors (i) via the addition of NHCs to ynals and subsequent oxidation of the Breslow intermediates (Figure 1B, I); (ii) via the reaction of NHCs with activated alkynoic acid esters (Figure 1B, II); (iii) via the reaction of NHCs with in situ activated alkynoic acids (Figure 1B, III). Typically, alkynyl acyl azoliums exhibit bielectrophilicity, and they have been investigated for [3 + n] annulations with diverse binucleophile reagents to afford heterocyclic molecules (Figure 1B, IV).

However, despite recent developments gradually enabling the diverse transformation of these species, the area is in the early stages of its development. The main challenge to exploring the reactivity of alkynyl acyl azoliums intermediates is attributed to the difficulty to control the chemo- and regioselectivities, which would result in undesired byproducts. For instance, the NHC-bound alkynyl acyl anion intermediates **Int. I**, allene intermediates **Int. II**, and alkenyl acyl azoliums intermediates **Int. III** might also be formed during the generation of alkynyl acyl azoliums intermediates (Figure 1C). Furthermore, the control of the regioselectivity of the [3 + n] annulations between binucleophile and alkynyl acyl azolium intermediates is another challenge to explore in this reaction (Figure 1C, IV,V).

Herein, we summarized the efforts and advances in the NHC-catalyzed [3 + n] annulation reactions involving alkynyl acyl azoliums intermediates with focus on their generation and reactivity as well as the mechanism of the reactions. We present these achievements in generally chronological order and some seminal efforts or closely related works are mentioned as well. All the NHCs described in this article are summarized in Figure 2.

## 2. Discussion

In 2018, Wang described the generation of alkynyl acyl azolium intermediates through the addition of NHC catalyst to ynals and subsequent oxidation of the Breslow intermediates (Figure 3) [26]. The reaction of alkynyl acyl azoliums with binucleophile cyclic 1,3-diones **1** affords the axially chiral *α*-pyrone-aryls **3**, along with byproducts **4**, **5** and **6** which are derived from the annulation of α,*β*-unsaturated acyl azoliums intermediate **Int. III**, regioselective annulation between alkynyl acyl azoliums intermediate and oxygen nucleophile of cyclic 1,3-diones **1**, as well as Knoevenagel reaction of **3** with **1**, respectively.

Mechanistically, the reaction proceeds via the addition of NHC catalyst to ynal **2** followed by oxidation of the Breslow intermediate to form NHC-bound alkynyl acyl azolium intermediate **7**. Nucleophilic addition of cyclic 1,3-diones **1** to **7 are** promoted by Lewis acid Mg(OTf)_2_ affords allenolate intermediate **9**. Subsequent proton transfer forms alkenyl acyl azolium intermediate **10**. Then, nucleophilic attack of acyl azolium forms an O—C bond and affords intermediate **11**. The release of NHC catalyst finally delivers the targeted product **3**. Importantly, the addition of Lewis acid was essential to modulate the regioselectivity. The chelation of the oxygen atom with magnesium ion promoted carbon nucleophilic addition of 1,3-dione to alkynyl acyl azoliums intermediate **7** and the formation of byproduct was inhibited. In addition, the attempt of oxidative dehydrogenation of **4** under their standard reaction conditions did not deliver product **3**, which suggested that the direct annulation of cyclic 1,3-dione with unsaturated acyl azolium intermediate pathway was excluded (Figure 4).

In 2020, Qi and co-workers developed a similar NHC-catalyzed [3 + 3] annulation of alkynyl acyl azoliums intermediate by replacing the binucleophile of pyrrol-4-one **13** [27]. In this reaction, the simple ynal **12a** underwent [3 + 3] annulation smoothly and afforded the non-axially chiral pyrones in good yield. By optimizing a particular class of chiral indanol-derived NHCs and other conditions, the formation of axially chiral pyrones was also proven to be feasible by using 3-(2-methoxynaphthalen-1-yl)propiolaldehyde (**12b**) in the presence of chiral NHC catalyst **A2** (Figure 5).

Chi, Jin and co-workers also disclosed a [3 + 3] annulation of NHC-bound alkynyl acyl azoliums with *N*-Ts imine **16** [24]. In this case, the alkynyl acyl azolium intermediate was generated by the reaction of NHC catalyst with activated alkynoic acid ester. The alkynyl acyl azolium intermediates have great potential for reaction discovery due to the highly reactive carbon–carbon triple bond. In this work, they explored the reactivity of alkynyl acyl azoliums with binucleophile *N*-Ts imine **16** in order to access a variety of functionalized pyridines **17** (Figure 6).

Mechanistically, the addition of NHC catalyst to the ester **15** affords the key alkynyl acyl azolium intermediate **19.** The nucleophilic conjugated addition of enamide **20** to **19** delivers the allenolate azolium intermediate **21**, which undergoes a proton transfer to form alkenyl acyl azolium intermediate **22.** Subsequent lactamization occurs to release the NHC catalyst and delivers the *N*-Ts *δ*-lactams **18**. Finally, isomerization of *N*-Ts *δ*-lactam **18** at a slightly elevated temperature produces the pyridines **17**. This work pioneered the use of activated alkynoic acid ester as the precursor to generate the alkynyl acyl azolium intermediate.

In 2020, Qi and co-workers explored even further the reactivity of alkynyl acyl azolium intermediate for the [3 + 3] annulation reaction (Figure 7) [28]. In this work, the *N*-Ts-protected 2-aminoacrylate **24** serves as a nucleophile for conjugated addition and affords a range of pyridines **25** in moderate yields.

A possible mechanism is proposed for the reaction. Addition of NHC catalyst to ynal **23** delivers Breslow intermediate **26,** then oxidation of **26** with DQ yields the alkynyl acyl azolium intermediate **27**. Subsequent 1,4-addition of *N*-Ts-protected 2-aminoacrylate **24** to **27** produces allenolate azolium intermediate **28**, which undergoes proton transfer and lactamization to afford *N*-Ts *δ*-lactam **30** and regenerates the NHC catalyst. Finally, thermodynamic aromatization achieves the product pyridines **25**. Interestingly, 1,2-addition of **24** to **27** and Claisen rearrangement with **31** pathways cannot be excluded.

Besides [3 + 3] annulation reactions, Du and co-workers also developed an NHC-catalyzed [3 + 2] annulation of alkynyl acyl azolium intermediate in 2017 [25]. In this process, activated esters **15** were used to generate the alkynyl acyl azolium intermediates, which reacted with *β*-diacyl **32** to afford the desired *Z*-2-vinylfuran-3(2*H*)-one **33** with various substituents in moderate to excellent yields. The reaction was compatible with both electron-withdrawing and electron-donating groups with regard to esters. However, undesired six-membered ring byproduct **39** was also observed with low to moderate yields in some cases (Figure 8).

Mechanistically, the reaction starts by nucleophilic addition of NHC catalyst to the ester **15** to afford the key alkynyl acyl azolium intermediate **34**. Deprotonation of *β*-diacyl **32** by DIPEA followed by complexation with LiCl yields acyl enolate **35**, which undergoes 1,2-addition to obtain **36**. Subsequent intramolecular proton transfer generates intermediate **37**. Two pathways may be involved in the intramolecular nucleophilic addition process. The 5-membered *Z*-2-vinylfuran-3(2*H*)-ones **33** are obtained when the addition of hydroxyl occurs at the α-carbon of the triple bond. On the other hand, 6*-endo-dig* cyclization of **37** yields the byproducts 4*H*-pyran-4-ones **39**. DFT calculations indicated that the *α*-carbons are positively charged and that the *β*-carbons were negatively charged in both intermediate **36** and **37**. Therefore, attack of the *α*-carbon by oxygen anion is more favorable for yielding five-membered products in the intramolecular addition process. Due to the less-steric hindrance between alkenyl hydrogen and carbonyl, *Z*-isomers are able to be obtained with high stereoselectivity (Figure 8).

To further investigate the reactivity of NHC-bound alkynyl acyl azoliums intermediates, Wang and co-workers extended binucleophile to amidines **40** [29]. In this case, they explored another NHC catalyzed [3 + 3] annulation of alkynyl acyl azoliums to construct multiply substituted pyrimidin-4-ones (Figure 9). The reaction was compatible with both electron-withdrawing and electron-donating groups on ynals and amidines. Furthermore, the desired products were obtained in good yields even with the bulkier amidines, which enabled this reaction to be applied for further diversification.

Mechanically, the catalytic cycle begins with the addition of NHC catalyst to ynal **23** to form Breslow intermediate **42**, which undergoes an oxidation to afford alkynyl acyl azolium intermediate **43**. The coordination of the Lewis acid Mg(OTf)_2_ with amidine **40** and intermediate **43** produces complex **44**. Then, Michael addition affords allenolate intermediate **45**, which undergoes intramolecular proton transfer to generate alkenyl acyl azolium intermediate **46;** subsequently, intramolecular 6-*exo*-*dig* cyclization delivers the final product, pyrimidin-4(1*H*)-one **41,** and regenerates the NHC catalyst.

In 2018, based on the works of NHC-catalyzed reactions with in situ activation of saturated or alkenoic acids [30,31,32,33,34,35], Du and co-workers reported the formation of alkynyl acyl azolium intermediate **50** via the in situ activation of alkynoic acids of **47** by NHC catalyst [36]. This seminal work achieved the NHC-catalyzed formal [3 + 3] annulation of alkynoic acids **47** and 2-mercaptoimidazoles **48** to access the heterocyclic imidazo[2,1-*b*][1,3]thiazinone frameworks (Figure 10).

Mechanistically, alkynoic acid **47** is activated in situ by pyBOP followed by the addition of NHC catalyst to afford alkynyl acyl azolium intermediate **50**. Michael addition of 2-mercaptoimidazoles **48** to intermediate **50** forms allenolate azolium intermediate **51**. Subsequent proton transfer produces alkenyl acyl azolium intermediate **52,** which undergoes 6-*exo*-*trig* cyclization to give formal [3 + 3] annulation product *δ*-lactam **49** with release of the NHC catalyst.

In 2019, based on previous works on the annulations of alkenyl acyl azolium intermediates with indolin-3-ones [37,38], Du and co-workers achieved [3 + 3] annulation of alkynyl acyl azolium intermediates with 3-oxo indolin-2-ides (Figure 11) [39]. In the presence of DBU and NHC iminium salt **A8**, 4-nitrophenyl alkynyl acid esters **15** and indolin-3-ones **53** underwent the [3 + 3] annulation smoothly and yielded pyrano[3,2-*b*]indol-2-ones **54** in an efficient and rapid manner. The benzofuran-3(2*H*)-one **60** was also explored as binucleophile for the [3 + 3] annulation reaction under standard conditions, which produce the corresponding 4-phenyl-2*H*-pyrano[3,2-*b*]benzofuran-2-one **61** product in 30% yield.

Mechanistically, the reaction is initiated by the addition of NHC catalyst to activated alkynoic acid esters **15** followed by the elimination of 4-nitrophenolate to afford alkynyl acyl azolium intermediate **56**. Michael addition of 3-oxo indolin-2-ide **59** to intermediate **56** forms allenolate azolium intermediate **57**. Subsequent proton transfer produces alkenyl acyl azolium intermediate **58** which undergoes 6-*exo*-*trig* cyclization to give corresponding formal [3 + 3] annulation product **54** with the release of the NHC free carbene. Although the nucleophilic addition of resonant isomer enolate **59′** of **59** to alkynyl acyl azolium intermediate **56** could produce byproduct **55**, its formation could be completely inhibited by optimizing the base and solvent. When the reaction carried out at 90 °C or tetrahydrofuran was used as the solvent, byproduct **55** was obtained in 30–45% yields. Reducing the heat from 90 °C to room temperature and replacing tetrahydrofuran with other solvent (toluene, acetonitrile, or dichloromethane) could essentially completely inhibit the formation of byproduct **55**, and the targeted product **54** could be obtained with high chemoselectivity.

Additionally, NHC-catalyzed [3 + 3] annulations of 4-nitrophenyl alkynyl acid esters **15** and benzofuran-3-amines **62** were also demonstrated by Du and co-workers to obtain functionalized benzofuro[3,2-*b*]pyridin-2-ones **63** (Figure 11) [40]. The reaction conditions are generally consistent, except for the binucleophile (**53**, **60** and **62**); the reaction mechanism is similar to the previous work. Deprotonation of benzofuran-3-amine **62** generates enamine ion **64**, which resonates with the 3-oxo indolin-2-ide **59** analogue 3-imino benzofuran-2-ide **64′**. 1,4-Conjugate addition of **64′** with alkynyl acyl azolium intermediate **56** and followed by lactamization affords the desired product **63**.

Based on the successful synthesis of achiral *δ*-lactones **54** or *δ*-lactams **63** via [3 + 3] annulations of alkynyl acyl azolium intermediate **56** with α-oxo ide **59** or α-imino ide **64′** [40], a related asymmetric annulation reaction was developed by Wei, Du, and co-workers in 2021 [41]. In the presence of potassium carbonate and NHC iminium salt **A9**, a steric hindrance alkynyl acid ester **66** activated by 4-nitrophenyl reacts with 2-sulfonamidoindolines **65** yielding axially chiral *δ*-lactam **68**, and subsequently thermodynamic aromatization produces an enantioenriched 4-aryl α-carboline **67** containing a chiral C–N axis (Figure 12).

Mechanistically, the reaction is initiated by the addition of NHC catalyst to activated alkynoic acid esters **66** followed by the elimination of 4-nitrophenolate to afford alkynyl acyl azolium intermediate **69**. Deprotonation of 2-sulfonamidoindolines **65** by potassium carbonate and followed by Michael addition to intermediate **69** forms allenolate azolium intermediate **70**. Subsequent proton transfer produces alkenyl acyl azolium intermediate **71** which undergoes lactamization to yield the corresponding formal [3 + 3] annulation *δ*-lactams **68** with release of the NHC catalyst. Further thermal treatment of *δ*-lactams **68** affords the aromatized product **67**. DFT calculation indicates that in the process of nucleophilic attack to intermediate **69**, the energy of transition state **TS1 *R*** is 1.8 kcal/mol lower than that of transition state **TS1 *S***, so *R* configuration isomer plays a dominant role in the reaction. According to DFT calculations, the main reason for the lower energy of **TS1 *R*** than **TS1 *S*** is that the noncovalent interactions (LP…*π*, C–H…N, *π*…*π*, etc.) of the former are stronger than the latter.

The above works referred to the construction of achiral compounds or molecules with the C–C axis through NHC-bound alkynyl acyl azolium intermediates. However, investigation of the C–hetero chiral axis remained underexplored. In 2021, Jin, Chi, and co-workers realized NHC-catalyzed asymmetric synthesis of C–N axial chiral thiazine **73** via the [3 + 3] annulation of alkynyl acyl azolium intermediate and thioureas **72** (Figure 13) [42]. In this approach, in the presence of NHC-free carbene **A10** and Scandium trifluoromethanesulfonate additive, a variety of bulky aryl substituted thioureas **72** annulated with ynals **23** afforded thiazine **73** with moderate to good yields and high to excellent enantioselectivities.

Mechanistically, the reaction proceeds via the addition of NHC catalyst to activated ynal **23a** in order to generate the Breslow intermediate **74**. The subsequent oxidation by DQ generates alkynyl acyl azolium intermediate **75**. Deprotonation of thiourea **23a,** through DMAP and nucleophilic thiol-addition to intermediate **75,** forms allenolate azolium intermediate **76**. Subsequent proton transfer produces alkenyl acyl azolium intermediate **77**, which complexes with Sc(OTf)_3_ to afford stereoisomeric intermediate **78**. Under the action of chiral NHC, intermediate **78** undergoes 6-*exo*-*trig* cyclization to yield the corresponding formal [3 + 3] annulation product **73a** with the release of the NHC catalyst. It is worth noting that although Scandium promotes the reaction, it has no effect on ee value.

In 2021, as a continuous work on NHC-catalyzed atroposelective [3 + n] annulation to access chiral C–N axis heterocyclic compounds, Chi and co-workers disclosed the atroposelective [3 + 2] annulation between ynals **23** and 4-arylurazole **79** using a desymmetrization strategy (Figure 14) [43]. A wide range of ynals **23** and symmetric urazole **79** with bulky 4-aryl bearing diverse substituents were well tolerated and underwent the desymmetric atroposelective [3 + 2] annulation to afford pyrazolo[1,2-*a*]triazoles **80** containing a C–N axis in good to excellent yield with high enantiomeric excess. The mechanism is similar to that reported previously, in which atroposelective nucleophilic addition of the deprotonated **79** to the alkynyl acyl azolium intermediate **75** occurs to yield formal [3 +2] annular product.

## 3. Conclusions

Since its discovery, NHC-based alkynyl acyl azolium intermediates have made significant progress in the past five years. Three methods to access the NHC-based alkynyl acyl azolium intermediates have been discussed, (1) via the oxidation of ynals, (2) via the activation of alkynoic acid ethers, and (3) via the in situ activation of alkynoic acids. These intermediates exhibit bielectrophilicity and react with binucleophiles via conjugated addition followed by 1,2-addition to yield structurally and functionally diverse cyclic molecules such as pyridines, lactones, and lactams containing a ring-fused structure, as well as multiple heteroatoms. Particularly, with the participation of alkynyl acyl azolium intermediates, the construction of C–N chiral axes and C–C chiral axes was achieved.

However, due to the formation of several other reactive species during the generation of alkynyl acyl azolium intermediates and the difficulty of controlling the regioselectivity of the nucleophilic addition, these reactions lead to the formation of several byproducts. Unfortunately, only a few reports concerning this topic have been produced. On the other hand, only four relatively successful cases for the synthesis of chiral compounds have been developed. In the future, more efforts will be required to explore new reactions for the synthesis of axially chiral molecules and to further investigate the mechanism by which NHC-based alkynyl acyl azolium intermediates participate.

## Data Availability

Not applicable.

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
