# Peer review of "A Review on Generation and Reactivity of the N-Heterocyclic Carbene-Bound Alkynyl Acyl Azolium Intermediates"

_molecules, 2022, doi:10.3390/molecules27227990_

Round 1

Reviewer 1 Report

This short review by Zhao et al summarized the history and application of NHC-bound alkynyl acyl azolium intermediates. This review is well-organized and easy to follow. The schemes and references are also appropriate. I would suggest to publish after minor revision:

(1) the authors can consider to add one more scheme in the introduction or conclusion part to list all of the NHC structures currently used in 3+n reactions. This will help readers to figure out the current choice of NHC structures.

(2) In scheme 1C, the authors summarized the possible challenge to explore the reactivity of NHC-alkynyl acyl azoliums (Int I-III). Which Int is more preferred in the reaction? Are these Ints always formed in the below mentioned references? 

Author Response

Responses to reviewer 1:

1.1) This short review by Zhao et al summarized the history and application of NHC-bound alkynyl acyl azolium intermediates. This review is well-organized and easy to follow. The schemes and references are also appropriate. I would suggest to publish after minor revision:

Our response:

We are grateful for the positive comments from this reviewer.

1.2) The authors can consider to add one more scheme in the introduction or conclusion part to list all of the NHC structures currently used in 3+n reactions. This will help readers to figure out the current choice of NHC structures.

Our response:

We have listed the NHC structures currently used in [3 + n] annulations in the introduction part (Scheme 2).

1.3) In scheme 1C, the authors summarized the possible challenge to explore the reactivity of NHC-alkynyl acyl azoliums (Int I-III). Which Int is more preferred in the reaction? Are these Ints always formed in the below mentioned references?

Our response:

Normally, Int I is more preferred. As we stated in the conclusions part “However, due to the formation of several other reactive species during the generation of alkynyl acyl azolium intermediates and the difficulty to control the regioselectivity of the nucleophilic addition, these reactions lead to the formation of several by-products. Unfortunately, only a few reports concerning this topic have been reported.” These Ints might be formed but rarely reported in the below mentioned references.

Reviewer 2 Report

Zhao et al. proposed a review concerning NHC-bounded alkynyl acyl azolium intermediates used in synthesis of achiral and axially chiral cyclic scaffolds via [m+n] annulations. This work sums up three different approaches that have been developed as well as the mechanisms involved in each of those. Generally speaking, three different parts have been clearly introduced with pertinent examples of the field as well as brief description of the involved catalytic cycle. 

I do recommend an overall revision of the text, which became sometimes complicated due to some extensively long sentences. Some grammatical incoherencies and typo mistakes are also present. I do recommend this work for publication in Molecules upon minor revision. Please, find below the suggestions and comments divided by section.

I suggest to replace the title "A Review of Generation and Reactivity of the N-Heterocyclic-Carbene-Bound Alkynyl Acyl Azolium Intermediates.  into" "A Review on Generation and Reactivity of the N-Heterocyclic-Carbene-Bound Alkynyl Acyl Azolium Intermediates”. 

I suggest, for better comprehension, a general reorganization of the chapters as follow: Chapter 1: 1.1 cycloaddition (general introduction) 1.2. [3+3] …1.3. [3+2] and so on. Either by the methods to access NHH-based alkynyl acyl azolium intermediates Also it should be better to describe the procedures in chronological order.

 As general remark, every time is described a proton transfer or the migration/shift of a functional group/moiety, this has to be highlighted also in the corresponding scheme (e.g. different colors or explicating it in other ways)

Abstract

-        Page 1 line 10 : used as an organocatalyst, preposition missing

-        Page 1 line 11 : non-umpolung instead of umplolung

-        Page 1 line 15 : this review, which covers instead of cover

-        Page 1 line 16 : advances in the synthesis instead of for the synthesis

Introduction

The introduction results a bit confusing and complicated to follow. I suggest to reformulate it in a clearer way, point by point, elucidating also the scope of the review. 

As general remark, every time is described a proton transfer 

-        Page 1 line 27 : complicated sentence with multiple mistakes, the following is suggested : …process, NHCs have been widely used as organometallic ligands as well as organocatalysts in itself, due to their extensive and diverse synthesis and versatility.

-        Page 1 line 32 : umpolung instead of umplolung

-        Page 1 line 34 : active species instead of specie, the singular has the same form

-        Page 1 line 35 : I suggest rather NHC-based catalysis over NHC catalysis

-        Page 1 line 38 : Complicated and incomprehensible sentence, Suggestion: …intermediates, initially identified independently by…between 2017 and 2018, were less commonly studied.

-        Page 1 line 42 : I suggest rather Until now instead of To date

-        Page 2 line 50 : The main challenges in order to explore challenges…

-        Page 2 line 53 : For instance instead of for instances

-        Page 2 line 57 : …another challenge to explore in this reaction

-        Page 2 line 62 : as well instead of forward

-        Page 2 Line 63 : In the Scheme 1C The title should be The main challenges to explore concerning the reactivity of alkynyl acyl azoliums

Discussion

-        Page 3 line 76: NHC-catalyzed, lowercase instead of uppercase

-        Page 4 line 81: intermediate instead of intermediates

-        Page 4 line 84: Then,….O-C bond and affords..

-        Page 4 line 85: the targeted product

-        Page 4 line 95: the simple ynal 12a

-        Page 4 line 98: Maybe its worth noting that we use a particular class of NHCs

-        Page 4 line 102: Use ‘in this case’ rather than ‘differently’

-        Page 5 line 111: lactams should be said instead of lactam as it is a general structure

-        Page 3 line 117: same as page 3 line 76

-        Page 5 line 118: the order of words should be the following:  ..co-workers explored even further the reactivity…

-        Page 5 line 122-123: several terms are not precise in this sentence, it is an addition of NHC free carbene that is activated by deprotonation of the corresponding iminium salt

-        Page 6 line 136 : to be compatible ‘with’ instead of to

-        Page 6 line 138 :  six-membered ring instead of six-member

-        Page 6 line 139 :  it would be interesting to justify in this case why the byproduct 39 is formed as a minor one, as well as the Zconfiguration of the obtained product, as it is clearly explained in the reference 25

-        Page 6 line 140 :  use rather ‘starts by’ over proceeds from, which has a meaning of source

-        Page 6 line 142 :  complexation instead of complexion

-        Page 7 line 149 :  same as page 3 line 76

-        Page 8 line 151 :  Magnesium triflate that is described in the mechanism as well as in the text, does not figure among reactives in the general reaction scheme above

-        Page 8 line 152 : same as page 3 line 76

-        Page 8 line 156 : again it is compatible ‘with both EWG and EDG groups…’

-        Page 8 line 158: the following formulation is suggested : the desired products were obtained in good yields even with bulkier amidines

-        Page 8 line 166: the sentence is a bit long and should be separated for better understanding by the lecturers

-        Page 8 line 173: alkynoic acids 47 are activated

-        Page 9 line 180: same as page 3 line 76

-        Page 9 line 185: DBU and NHC iminium salt instead of pre-NHC

-        Page 9 line 195:  NHC free carbene instead of catalyst

-        Page 9 line 198:  The optimization of the base and solvent should be briefly discussed as it enables byproduct-free reaction

-        Page 10 line 207 : The solvent naming is inconsistent with the previous one, I suggest to use DCM or CH2Cl2 everywhere, time of the reaction should be separated: 2 h instead of 2h

-        Page 10 line 210 : Based on instead of on the basis on

-        Page 10 line 212 : has been developed instead of had been

-        Page 10 line 213 : The following formulation is suggested : …and  NHC iminium salt.. alkynyl acid ester activated by steric hindrance ..

-        Page 10 line 216 : produces an enantioenriched …containing a chiral C-N axis

-        Page 11 line 225: the transition state TS1 R is favored because it is lower in energy not higher, as you have described in the scheme 11, and further explanation should be given to this

-        Page 11 line 225-226: R and S absolute configuration should be in italic

-        Page 11 line 227 : in the scheme , the catalyst should appear clearly in the reaction scheme, not only in the mechanistic part

-        Page 11 line 232 : remained instead of has remained as it is a finished event

-        Page 11 line 235 : I suggest the following: …in presence of NHC free carbene and Scandium triflate additive…

-        Page 12 line 237: afforded instead of infinitive

-        Page 12 line 248: In the scheme the H+ transfer should be renamed in proton transfer to be consistent with previous schemes

-        Page 12 line 249: same as page 3 line 76

-        Page 12 line 250: ‘as a continuous work on’ is suggested

-        Page 12 line 255: triazoles instead of triazole

-        Page 12 line 258: …intermediate 75 occurs to yield formal…

-        Page 13 line 260: same as page 3 line 76

-        Page 13 line 264: discussed instead of disclosed

-        Page 13 line 270: the construction… was achieved...

-        Page 13 line 271: this long sentence should be reformulated, I suggest the following/ However, due to the formation of several other reactive species during the generation of alkynyl acyl azolium intermediates and the difficulty to control the regioselectivity of the nucleophilic addition, these reactions lead to a formation of several byproducts. 

-        Page 13 line 273: the following is suggested: Unfortunately, only a few reports concerning this topic have been reported

Page 13 line 277-278: … the mechanism, by which….intermediates participate

Author Response

Responses to reviewer 2:

2.1) Zhao et al. proposed a review concerning NHC-bounded alkynyl acyl azolium intermediates used in synthesis of achiral and axially chiral cyclic scaffolds via [m + n] annulations. This work sums up three different approaches that have been developed as well as the mechanisms involved in each of those. Generally speaking, three different parts have been clearly introduced with pertinent examples of the field as well as brief description of the involved catalytic cycle.

I do recommend an overall revision of the text, which became sometimes complicated due to some extensively long sentences. Some grammatical incoherencies and typo mistakes are also present. I do recommend this work for publication in Molecules upon minor revision. Please, find below the suggestions and comments divided by section.

Our response:

Thanks for the comments from this reviewer which help us to improve the manuscript. We have revised the manuscript carefully based on your suggestions.

2.2) I suggest to replace the title "A Review of Generation and Reactivity of the N-Heterocyclic-Carbene-Bound Alkynyl Acyl Azolium Intermediates”.  into" "A Review on Generation and Reactivity of the N-Heterocyclic-Carbene-Bound Alkynyl Acyl Azolium Intermediates”.

Our response:

We have replaced “of” into “on” in the revised manuscript.

2.3) I suggest, for better comprehension, a general reorganization of the chapters as follow: Chapter 1: 1.1 cycloaddition (general introduction) 1.2. [3+3] …1.3. [3+2] and so on. Either by the methods to access NHC-based alkynyl acyl azolium intermediates Also it should be better to describe the procedures in chronological order.

Our response:

Thanks for the suggestions. We do consider to organize the chapters as this reviewer’s suggestion during the preparation of the manuscript. However, since there are only 12 related works, each chapter will be even thinner if subdivided as mentioned. As suggested by the reviewers, it may be helpful to divide the discussion into [3+3], [3+2] and so on. However, 10 works are involved to [3+3] annulations while only 2 works are related to [3+2] annulations. If divided in this way, it seems that the former part has too many contents.

In addition, we have also considered to divide the manuscript into three sections according to the reaction precursor. However, because there is only one work to generate the intermediate via in situ activation of alkynoic acid act, the contents of this separate section would appear too few and abrupt.

2.4) As general remark, every time is described a proton transfer or the migration/shift of a functional group/moiety, this has to be highlighted also in the corresponding scheme (e.g. different colors or explicating it in other ways)

Our response:

We have used different colors to highlight these chemical changes in the corresponding schemes.

2.5) Abstract

- Page 1 line 10: used as an organocatalyst, preposition missing

- Page 1 line 11: non-umpolung instead of umplolung

- Page 1 line 15: this review, which covers instead of cover

- Page 1 line 16: advances in the synthesis instead of for the synthesis

Our response:

We have corrected these grammatical incoherencies and typo mistakes.

2.6) Introduction

The introduction results a bit confusing and complicated to follow. I suggest to reformulate it in a clearer way, point by point, elucidating also the scope of the review.

Our response:

We have reformulated the introduction into four relatively separate paragraphs in the revised manuscript, making the scope of the elaboration more explicit.

2.7) Introduction

- Page 1 line 27: complicated sentence with multiple mistakes, the following is suggested : …process, NHCs have been widely used as organometallic ligands as well as organocatalysts in itself, due to their extensive and diverse synthesis and versatility.

- Page 1 line 32: umpolung instead of umplolung

- Page 1 line 34: active species instead of specie, the singular has the same form

- Page 1 line 35: I suggest rather NHC-based catalysis over NHC catalysis

- Page 1 line 38: Complicated and incomprehensible sentence, Suggestion: …intermediates, initially identified independently by…between 2017 and 2018, were less commonly studied.

- Page 1 line 42: I suggest rather Until now instead of To date

- Page 2 line 50: The main challenges in order to explore challenges…

- Page 2 line 53: For instance instead of for instances

- Page 2 line 57: …another challenge to explore in this reaction

- Page 2 line 62: as well instead of forward

- Page 2 Line 63: In the Scheme 1C The title should be The main challenges to explore concerning the reactivity of alkynyl acyl azoliums

Our response:

We have corrected these errors and made corresponding modifications as suggested.

2.8) Discussion

- Page 4 line 98: Maybe its worth noting that we use a particular class of NHCs.

- Page 6 line 139: it would be interesting to justify in this case why the byproduct 39 is formed as a minor one, as well as the Z-configuration of the obtained product, as it is clearly explained in the reference 25.

- Page 9 line 198: The optimization of the base and solvent should be briefly discussed as it enables byproduct-free reaction.

- Page 11 line 225: the transition state TS1 R is favored because it is lower in energy not higher, as you have described in the scheme 11, and further explanation should be given to this.

Our response:

Relevant descriptions have been added on page 4 line 101–102, page 7 line 152–157, page 10 line 210–213, and page 12 line 243–245 in the revised manuscript.

2.9) Discussion

- Page 3 line 76: NHC-catalyzed, lowercase instead of uppercase

- Page 4 line 81: intermediate instead of intermediates

- Page 4 line 84: Then,….O-C bond and affords..

- Page 4 line 85: the targeted product

- Page 4 line 95: the simple ynal 12a

- Page 4 line 102: Use ‘in this case’ rather than ‘differently’

- Page 5 line 111: lactams should be said instead of lactam as it is a general structure

- Page 5 line 117: same as page 3 line 76

- Page 5 line 118: the order of words should be the following:  ..co-workers explored even further the reactivity…

- Page 5 line 122-123: several terms are not precise in this sentence, it is an addition of NHC free carbene that is activated by deprotonation of the corresponding iminium salt

- Page 6 line 136: to be compatible ‘with’ instead of to

- Page 6 line 138: six-membered ring instead of six-member

- Page 6 line 140: use rather ‘starts by’ over proceeds from, which has a meaning of source

- Page 6 line 142: complexation instead of complexion

- Page 7 line 149: same as page 3 line 76

- Page 8 line 151: Magnesium triflate that is described in the mechanism as well as in the text, does not figure among reactives in the general reaction scheme above

- Page 8 line 152: same as page 3 line 76

- Page 8 line 156: again it is compatible ‘with both EWG and EDG groups…’

- Page 8 line 158: the following formulation is suggested : the desired products were obtained in good yields even with bulkier amidines

- Page 8 line 166: the sentence is a bit long and should be separated for better understanding by the lecturers

- Page 8 line 173: alkynoic acids 47 are activated

- Page 9 line 180: same as page 3 line 76

- Page 9 line 185: DBU and NHC iminium salt instead of pre-NHC

- Page 9 line 195: NHC free carbene instead of catalyst

- Page 10 line 207: The solvent naming is inconsistent with the previous one, I suggest to use DCM or CH2Cl2 everywhere, time of the reaction should be separated: 2 h instead of 2h

- Page 10 line 210: Based on instead of on the basis on

- Page 10 line 212: has been developed instead of had been

- Page 10 line 213: The following formulation is suggested: …and  NHC iminium salt.. alkynyl acid ester activated by steric hindrance.

- Page 10 line 216: produces an enantioenriched …containing a chiral C-N axis

- Page 11 line 225-226: R and S absolute configuration should be in italic

- Page 11 line 227: in the scheme , the catalyst should appear clearly in the reaction scheme, not only in the mechanistic part

- Page 11 line 232: remained instead of has remained as it is a finished event

- Page 11 line 235: I suggest the following: …in presence of NHC free carbene and Scandium triflate additive…

- Page 12 line 237: afforded instead of infinitive

- Page 12 line 248: In the scheme the H+ transfer should be renamed in proton transfer to be consistent with previous schemes

- Page 12 line 249: same as page 3 line 76

- Page 12 line 250: ‘as a continuous work on’ is suggested

- Page 12 line 255: triazoles instead of triazole

- Page 12 line 258: …intermediate 75 occurs to yield formal…

- Page 13 line 260: same as page 3 line 76

Our response:

We have corrected these errors and made corresponding modifications as suggested.

2.10) Conclusions

- Page 13 line 264: discussed instead of disclosed

- Page 13 line 270: the construction… was achieved...

- Page 13 line 271: this long sentence should be reformulated, I suggest the following/ However, due to the formation of several other reactive species during the generation of alkynyl acyl azolium intermediates and the difficulty to control the regioselectivity of the nucleophilic addition, these reactions lead to a formation of several byproducts.

- Page 13 line 273: the following is suggested: Unfortunately, only a few reports concerning this topic have been reported

Page 13 line 277-278: … the mechanism, by which….intermediates participate

Our response:

We have corrected these errors and made corresponding modifications as suggested.
